# Renin-Angiotensin-System Inhibitors for the Prevention of Chemotherapy-Induced Peripheral Neuropathy: OncoToxSRA, a Preliminary Cohort Study

**DOI:** 10.3390/jcm11102939

**Published:** 2022-05-23

**Authors:** Simon Frachet, Aurore Danigo, Marc Labriffe, Flavien Bessaguet, Bianca Quinchard, Nicolas Deny, Kim-Arthur Baffert, Elise Deluche, Franck Sturtz, Claire Demiot, Laurent Magy

**Affiliations:** 1Department of Neurology, Reference Center for Rare Peripheral Neuropathies, University Hospital of Limoges, 87000 Limoges, France; laurent.magy@chu-limoges.fr; 2UR 20218-NeurIT, Faculties of Medicine and Pharmacy, University of Limoges, 87025 Limoges, France; aurore.danigo@unilim.fr (A.D.); bianca.quinchard@gmail.com (B.Q.); franck.sturtz@unilim.fr (F.S.); claire.demiot@unilim.fr (C.D.); 3Department of Pharmacology, Toxicology and Pharmacovigilance, University Hospital of Limoges, 87000 Limoges, France; marc.labriffe@unilim.fr; 4Pharmacology & Transplantation, INSERM U1248, University of Limoges, 87025 Limoges, France; 5INSERM 1083 CNRS UMR 6015 Mitovasc Laboratory, CarMe Team, University of Angers, 49045 Angers, France; flavien.bessaguet@univ-angers.fr; 6Department of Medical Oncology, University Hospital of Limoges, 87000 Limoges, France; nicolas.deny@chu-limoges.fr (N.D.); kim-arthur.baffert@unilim.fr (K.-A.B.); elise.deluche@chu-limoges.fr (E.D.); 7Department of Biochemistry and Molecular Genetics, University Hospital of Limoges, 87000 Limoges, France

**Keywords:** chemotherapy-induced peripheral neuropathy, renin-angiotensin system inhibitor, neuroprotection, pain, electrochemical skin conductance

## Abstract

Chemotherapy-induced peripheral neuropathy (CIPN) is a frequent and dose-limiting adverse side effect of treatment. CIPN affects the oncological prognosis of patients, as well as their quality of life. To date, no specific pharmacological therapy has demonstrated effectiveness in preventing CIPN. Accumulating preclinical evidence suggests that renin-angiotensin system (RAS) inhibitors may have neuroprotective effects. One hundred and twenty patients were included in this observational study and were followed from the beginning of their neurotoxic chemotherapy schedule until their final assessment, at least one month after its cessation. The National Cancer Institute’s common toxicity criteria 4.0 (NCI-CTC 4.0) were used to grade the severity of adverse events. Follow-ups also included electrochemical skin conductance and scales for pain, quality of life and disability. Among patients receiving a platinum-based regimen, the mean grade of sensory neuropathy (NCI-CTC 4.0) was significantly lower in the RAS inhibitor group after the end of their anticancer treatment schedule. Because of the observational design of the study, patients in the RAS inhibitor group cumulated comorbidities at risk of developing CIPN. Randomized controlled trials in platinum-based regimens would be worth conducting in the future to confirm the neuroprotective potential of RAS inhibitors during chemotherapy.

## 1. Introduction

Chemotherapy-induced peripheral neuropathy (CIPN) is a frequent and major dose-limiting side effect of several anticancer agents [1]. CIPN usually presents as predominantly painful sensory symptoms and instances of “stocking and glove” distribution that may persist after the cessation of chemotherapy. CIPN is agent-dependent and is particularly frequent in platinum- and taxane-based regimens [2,3]. To date, no pharmacologic agent has demonstrated a significant neuroprotective effect in patients suffering from CIPN [4]. The only currently recognized prevention is based on dose reduction or withdrawal of the neurotoxic drug after the early recognition of neuropathic symptoms. CIPN, thus, impacts both the quality of life of patients and their oncological prognosis, due to residual pain and necessary changes to the treatment regimen. With an incidence of nearly 25 million cancer cases worldwide [5] and given that new anticancer agents still induce CIPN, it is becoming a matter of urgency to develop neuroprotective strategies [6].

The renin-angiotensin system (RAS) plays a major role in homeostasis, including the control of blood pressure and fluid balance. Initially described as an endocrine-only system, local RAS is now known to be widely expressed in many tissues [7], notably in the sensory peripheral nervous system [8]. Some preclinical data has revealed that RAS inhibitors, such as angiotensin-converting enzyme (ACE) inhibitors and angiotensin receptor blockers (ARB), exert a neuroprotective effect in various murine models of peripheral neuropathy, potentially via angiotensin II type 2 receptor stimulation [9,10]. These effects have been replicated in both traumatic [11], diabetic [12] and toxin-induced [13] neuropathies. A neuroprotective effect of RAS inhibitors was also demonstrated in animal models of chemotherapy-induced neuropathy: vincristine-induced neuropathy, treated by candesartan [14]; oxaliplatin-induced neuropathy, treated with ramipril in mice [15]; and in a rat model of paclitaxel-induced neuropathy, treated with losartan [16]. In these models, RAS inhibitors alleviated the neuropathic pain and nerve injury associated with chemotherapy.

In humans, Malik et al. explored the potential beneficial effect of RAS antagonists in the context of diabetic peripheral neuropathy. In a double-blinded, placebo-controlled trial, the ACE inhibitor, trandolapril, improved peroneal nerve conduction velocity in patients with diabetes [17]. More recently, a retrospective study compared the quantitative sensory testing values of patients taking RAS inhibitors before neurotoxic chemotherapy exposure vs. patients who did not receive RAS inhibitors [18]. The results suggested that RAS inhibitor treatment offered a protective effect on sensory nerve fiber function when evaluated by cold-pain detection and touch-detection thresholds. Another retrospective cohort study suggested that patients already treated with RAS inhibitors were slightly less likely to develop neurotoxicity during oxaliplatin treatment [19].

In the present study, our goal was to prospectively examine the neuroprotective potential of RAS inhibitors in chemotherapy-induced neurotoxicity. We hypothesized that RAS inhibitor treatment would be neuroprotective during chemotherapy and would result in lower grades of CIPN compared with controls. As a consequence, we expected that patients in the RAS inhibitor group would require the discontinuation of chemotherapy due to neurotoxicity less frequently and have better reported outcomes for pain, quality of life and disability.

## 2. Materials and Methods

### 2.1. Study Design and Participants

OncoToxSRA was a prospective, observational, monocentric cohort study performed in the Oncology Department of the University Hospital of Limoges, France.

Consecutive patients with solid cancers, who were about to receive chemotherapy known to have neurotoxic side effects for the first time, were enrolled in the study (excepting patients with lung cancer, who were locally treated in a separate clinic). Participants were required to be older than 18 years and able to give informed, signed consent to participate in the study. They also had to have no known pre-existing neuropathy before anticancer agent administration.

All patients were registered in the study on the day that they received their first dose of chemotherapy, just prior to its administration. The protocol consisted of an initial evaluation, while follow-up was on a quarterly basis until the final evaluation at least one month after the end of the neurotoxic chemotherapy regimen. These assessments took place during scheduled appointments, as part of the routine follow-up. An assessor, independent of the oncologists who managed the patients, conducted the CIPN evaluation. Participation in the study did not alter the medical management of the patient’s cancer.

The concomitant use of RAS inhibitors with chemotherapy was systematically recorded, with details of the international nonproprietary name and dosage. Thus, two groups were defined, based on whether RAS inhibitors were administered or not: the “RAS inhibitor” and “no RAS inhibitor” groups.

### 2.2. Outcomes

The primary endpoint was the severity of sensory neuropathy, scored according to the National Cancer Institute’s common toxicity criteria 4.0 (NCI-CTC 4.0) at the last follow-up when the cumulative dose of neurotoxic chemotherapy was at maximum. This scale ranks the severity of sensory neuropathy from 0 to 4 and is the instrument most commonly used in the assessment of CIPN [20]. This final follow-up needed to be performed at least one month after the end of the neurotoxic chemotherapy.

Secondary outcome measures included pain, disability, quality of life and sudomotor function at the last follow-up.

The neuropathic pain symptom inventory (NPSI) [21] was used to evaluate neuropathic pain; this is a self-administered questionnaire completed by the patient that includes 10 descriptors, quantified on a numerical scale from 0 to 10.

Disability was assessed using the Rasch-built overall disability scale for patients with CIPN (CIPN-RODS) [22], a 28-item scale of daily living activities, converted into a centile metric ability score ranging from 0 (lowest) to 100 (highest).

Quality of life was quantified by the 5-level EuroQol-5D (EQ5D5L), a 5-dimension scale ranging from 5 (best) to 25 (worst).

Sudomotor function was quantitated using a Sudoscan^®^ (Impeto Medical, Paris, France) through the non-invasive measurement of the electrochemical skin conductance (ESC) of the hands and feet. Previous evaluation of this device suggested that it was effective in the screening and follow-up of chemotherapy-related small fiber neuropathy [23]. The sudomotor function values were analyzed as a quantitative variable, averaged across the hands and feet.

A composite criterion of neurotoxicity was also developed in a secondary analysis and was defined as a discontinuation or dose reduction of chemotherapy due to neurotoxicity, as recorded in the patient’s medical file.

### 2.3. Statistical Methods

Descriptive statistics were used to examine the baseline characteristics and outcomes between groups of treatment (RAS or no RAS inhibitors) and according to the class of chemotherapeutic agent (platinum-based and Taxane). Analyses were conducted with Fisher’s exact test or the chi-squared test for qualitative variables, and the Mann–Whitney U-test or Student’s *t*-test for quantitative variables, as appropriate.

As had been previously reported [19], given the fact that oxaliplatin was the far most frequently prescribed neurotoxic chemotherapy treatment in the present cohort, we included all follow-ups of oxaliplatin-treated patients to estimate the cumulative dose of oxaliplatin to an event: (i) the onset of at least grade 2 neuropathy (NCI-CTC 4.0); or (ii) our composite criterion of neurotoxicity (as described above). Cumulative incidence curves were used to estimate the dose-to-event rates, using the Kaplan–Meier method and comparison by log-rank test. Subsequently, crude hazard ratios (HR) between groups were obtained using a univariate Cox proportional hazards model analysis to compare the risk factors associated with a cumulative dose of oxaliplatin to the onset of events.

Statistical analyses were performed using RStudio software (version 1.4.1103, RStudio, PBC, Boston, MA, USA, URL: http://www.rstudio.com/) and GraphPad Prism 8 (San Diego, CA, USA).

## 3. Results

### 3.1. Patients

One hundred and twenty patients were recruited and followed up from March 2019 to October 2020. At recruitment, 33 patients were already receiving an RAS inhibitor for a cardiovascular condition (RAS inhibitor group) and 87 were not (no RAS inhibitor group). In all, 17 patients were lost during follow-up; thus, 85.8% of all recruited patients reached the final follow-up, one month after the end of neurotoxic chemotherapy. The main reason for patients being lost to follow-up was death (14 out of 17 patients). Figure 1 shows the flow diagram of the study.

The characteristics of the tumors and chemotherapies used were balanced between the RAS inhibitor and the no RAS inhibitor groups (Table 1). In the RAS inhibitor group, patients were significantly older, had a significantly higher body mass index and were more frequently affected by hypertension. The proportion of diabetic patients was higher in the RAS inhibitor group; this difference was not statistically significant. At baseline, two of these diabetic patients in the RAS inhibitor group were found to have minor neuropathic symptoms. This accounted for a statistically significant difference between the RAS and the no RAS groups in the NCI-CTC 4.0 at baseline. Since their neuropathy was minor and not previously recorded, these patients were not excluded from the study. It is of note that ESC was comparable between the groups. Disability was significantly greater in the RAS inhibitor group, according to the CIPN-RODS.

The RAS inhibitors prescribed and their daily doses are summarized in Table 2. Equal proportions of patients were taking ARB and ACE inhibitors. The most commonly used ARB was irbesartan (69%), while the most commonly used ACE inhibitor was perindopril (41%).

### 3.2. Outcomes

Table 3 shows the comparisons after the final follow-up between patients with or without a RAS inhibitor, according to their class of neurotoxic chemotherapy.

Concerning our primary outcome, the mean grade of sensory neuropathy according to the NCI-CTC 4.0 was 0.4 in the RAS inhibitor group and 0.7 in the group without RAS inhibitors, among patients receiving a platinum-based chemotherapy treatment (*p* = 0.047). No statistically significant difference was found in patients receiving a taxane-based regimen (*p* = 0.853).

Concerning our secondary outcomes, no statistically significant difference was found between the RAS and no RAS groups for NPSI, EQ5D5L and ESC among patients receiving platinum-based or taxanes chemotherapy. Disability, as measured by CIPN-RODS, was more pronounced in the RAS inhibitor group at baseline (Table 1) and was comparable between the two groups at the last follow-up, whatever the class of chemotherapy (Table 3).

### 3.3. Oxaliplatin Cumulative Dose to Event

Figure 2 displays the cumulative dose of oxaliplatin to the composite criterion of neurotoxicity and the onset of grade 2 peripheral neuropathy, according to NCI-CTC 4.0. Despite apparent differences in the curves, no significant difference was found between the groups for the composite criterion (*p* = 0.12) nor in the onset of peripheral neuropathy of grade 2 or more (*p* = 0.18). Univariate hazard ratios (HR) were 0.44 (CI 0.18–1.07) and 0.37 (CI 0.12–1.19), respectively.

## 4. Discussion

To our knowledge, this is the first prospective study to evaluate the putative effect of RAS inhibitors, prescribed for their usual indications, to prevent the development or reduce the severity of CIPN. The strength of our study is in the long follow-up carried out throughout the chemotherapy schedule, with a detailed assessment of multiple indicators, including the grade of neuropathy, neuropathic pain, disability, quality of life and ESC.

Based on our primary outcome, the grade of neuropathy according to NCI-CTC 4.0, we found a significantly lower severity of neuropathy in the RAS inhibitor group among patients receiving platinum-based chemotherapy (Table 3). Although there was a slight tendency in this group for better parameters in terms of pain, quality of life and ESC with a RAS inhibitor, we did not find a statistically significant difference for all our secondary outcomes. Among patients receiving taxane-based chemotherapy, we did not find any difference between the groups for all considered parameters. However, the pathophysiology of CIPN is agent- and dose-dependent and is responsible for various clinical pictures. Indeed, according to the different agents and the doses administered, patients may develop length-dependent axonal neuropathy or, more rarely, sensory neuronopathy (ganglionopathy) [24].

In this way, since oxaliplatin was the far most widely prescribed neurotoxic chemotherapy in our cohort, we also performed a secondary analysis among these patients to assess whether there was a difference between groups, in terms of cumulative dose, to the onset of at least grade 2 neuropathy (NCI-CTC 4.0) or to our composite criterion of neurotoxicity. We did not find a statistically significant neuroprotective effect, despite a tendency to a protective effect, with the hazard ratio being 0.44 (95% CI: 0.18–1.07) and 0.37 (95% CI: 0.12–1.19), respectively.

Because of the non-interventional design of the study, patients receiving a RAS inhibitor at baseline had more risk factors for developing CIPN, such as a higher body mass index [25], greater disability and older age [26]. RAS inhibitors are commonly used in the treatment of hypertension [27]. As hypertension increases with age, it appears evident that the patients in the RAS inhibitor group were older, and so were likely to be more disabled and with a higher BMI. There was also a trend toward a higher incidence of diabetes [28] in the RAS inhibitor group. In fact, hypertension is an extremely common problem in patients with diabetes [29] and is known to be a risk factor for developing neuropathic symptoms [30]. Moreover, two of the diabetic patients in the RAS inhibitor group were discovered at baseline to have previously unrecorded minor neuropathic symptoms. The inherent comorbidities of the RAS inhibitor population have to be taken into account in the interpretation of the results because they may have diluted the neuroprotective effect of RAS inhibitors.

The current management of CIPN is limited to detecting neuropathic symptoms as early as possible, in order to reduce or stop neurotoxic chemotherapy and allow a partial or complete reversal of those symptoms. The consequence is a decrease in the observable neuroprotective effect of the candidate treatment. For this reason, we developed a composite criterion of neurotoxicity among patients receiving oxaliplatin in our secondary analysis, which allowed the detection by the oncologist of events that heralded chemotherapy-induced neurotoxicity. This composite criterion of neurotoxicity data suggested a tendency for lesser modification of chemotherapy protocols due to neurotoxicity in the RAS inhibitor group, albeit not a significant one.

The results from this exploratory study cannot assert the neuroprotective effect of RAS inhibitor treatment against the onset of CIPN. However, the observed suggestion of a neuroprotective effect of RAS inhibitors among patients receiving platinum-based chemotherapy attracted our interest. In the absence of therapeutic alternatives, there is an urgent medical need to identify neuroprotective medications. Since RAS inhibitors are widely prescribed and well-tolerated medications, these data should provide encouragement for future therapeutic trials. It is likely that this neuroprotective effect, if confirmed, could be more marked in the case of specific chemotherapy drugs, particularly RAS inhibitor molecules, dosages and combinations. Moreover, even we are focused on chemotherapy-induced neurotoxicity, it would be important to notice that cardiovascular adverse effects, such as hypertension, could be associated with chemotherapy administration at short- and long-term [31]. In this case, the use of RAS inhibitors would counteract both vascular and neuro-toxicities. Since all RAS inhibitors were pooled by necessity in the present observational study, the study may have been insufficiently powered to distinguish the possible combinatory effects. Randomized clinical trials, in addition to correcting issues of the comparability of groups, would make it possible to focus on the most promising neuroprotective combinations. Based on our observations, platinum-based regimens could form the first line of investigation.

## Figures and Tables

**Figure 1 jcm-11-02939-f001:**
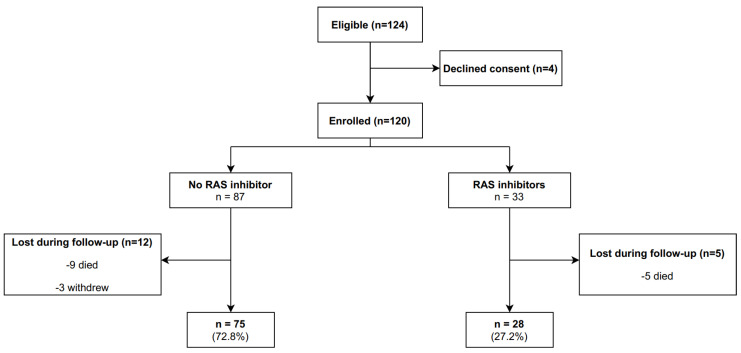
Flow diagram describing the recruitment and progress of participants in the cohort study. RAS: renin-angiotensin system.

**Figure 2 jcm-11-02939-f002:**
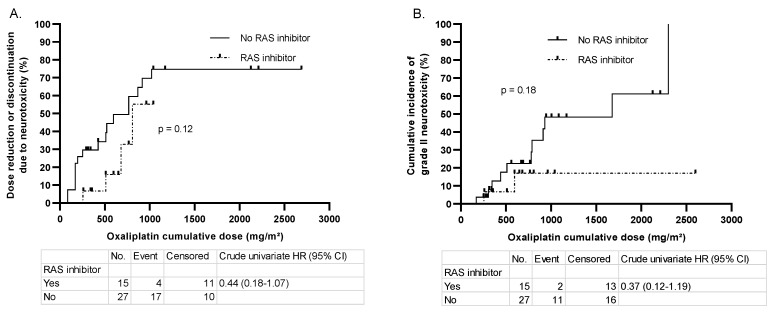
Kaplan–Meier curves and univariate Cox proportional hazard analysis, comparing the cumulative dose of oxaliplatin with the composite criterion of neurotoxicity (**A**) and grade 2 NCI-CTC 4.0 peripheral neuropathy (**B**). Patients who did not develop the endpoint after the completion of chemotherapy were plotted on the curves. The *p*-values were calculated using the log-rank test. CI: confidence interval; HR: hazard ratio; RAS: renin-angiotensin system.

**Table 1 jcm-11-02939-t001:** Characteristics of patients at study entry, expressed as median and interquartile ranges.

Variables	No RAS Inhibitor (*n* = 87)	RAS Inhibitor (*n* = 33)	*p*-Values
Patients			
Age in years median (range)	63 (25–86)	69 (49–81)	0.024 *
Gender (M/F) (%)	49/51	58/42	0.553
Body mass index (kg/m²)	24	28.7	0.004 **
Diabetes mellitus (%)	6	18	0.069
Alcohol consumption (%)	26	24	0.991
Hypertension (%)	26	100	<0.001 ***
Performance Status	0 (0–2)	0 (0–2)	0.221
**Cancer**			
**Localization**			0.317
Gastrointestinal (%)	38	55	
Gynecologic (%)	32	27	
ORL (%)	18	15	
Urogenital (%)	12	3	
**Classification/Line of therapy**			0.907
Adjuvant (%)	56	58	
Metastatic (%)	35	36	
Neoadjuvant (%)	9	6	
**Chemotherapy and cumulative dose**			
**Platinum-based (%)**	68.9	75.9	0.661
Carboplatin (%)	4.6	15.2	0.112
Cumulative dose (AUC)	30 (23.75–33.12)	30 (30–30)	0.893
Cisplatin (%)	31	15.2	0.105
Cumulative dose (mg/m²)	660 (562.71–900)	473.6 (300–675)	0.183
Oxaliplatin (%)	33.3	45.5	0.289
Cumulative dose (mg/m²)	916.5 (429.6–1152)	595 (347.8–786)	0.125
**Taxane-based (%)**	37.8	30.4	0.670
Cabazitaxel (%)	1.1	0	1.000
Cumulative dose (mg/m²)	250 (250–250)	NA	NA
Docetaxel (%)	26.4	15.2	0.233
Cumulative dose (mg/m²)	400 (252.1–425.6)	300 (225.5–400)	0.880
Paclitaxel (%)	10.3	15.2	0.527
Cumulative dose (mg/m²)	1142.3 (465.6–1840)	783.8 (700–1304.2)	0.519
**Neurotoxic association (%)**	10.3	15.2	0.344
**Neurotoxic chemotherapy duration (days)**	107 (45–156)	95.5 (62.3–137)	0.785
**Scores at baseline**			
**NCI-CTC 4.0 grade (0–4)**	0 (0–0)	0 (0–0)	
mean	0	0.061	0.022 *
**NPSI (0–100)**	0 (0–0)	0 (0–0)	0.195
**CIPN-RODS (0–100)**	84 (73–100)	73 (67–87)	0.010 *
**EQ5D5L (5–25)**	7 (5.5–9)	9 (7–10)	0.066
**ESC (µS)**			
Hands	71 (56–81)	65 (51–80)	0.486
Feet	77 (66–82)	74 (54–83)	0.525

Quantitative variables are expressed as median (interquartile range), except age as median (min-max). *p*-values were calculated with Fisher’s exact test or the chi-squared test for qualitative variables and the Mann-Whitney U-test or Student’s *t*-test for quantitative variables, as appropriate. * *p* < 0.5, ** *p* < 0.01 and *** *p* < 0.001 RAS inhibitor vs. no RAS inhibitor groups. CIPN-RODS: Rasch-built overall disability scale for patients with chemotherapy-induced peripheral neuropathy; EQ5D5L: 5-level EuroQol Research Foundation questionnaire; ESC: electrochemical skin conductance; NCI-CTC 4.0: National Cancer Institute common toxicity criteria; NPSI: neuropathic pain symptom inventory; RAS: renin-angiotensin system.

**Table 2 jcm-11-02939-t002:** RAS inhibitors used and the daily dose (median (min-max)).

RAS Inhibitors	Daily Dose (mg)
**ARB (*n* = 16)**	
Irbesartan (*n* = 11)	150 (150–300)
Candesartan (*n* = 3)	8 (4–16)
Losartan (*n* = 1)	50
Telmisartan (*n* = 1)	80
**ACE inhibitor (*n* = 17)**	
Perindopril (*n* = 7)	4.5 (2.5–10)
Ramipril (*n* = 6)	5 (1.25–5)
Enalapril (*n* = 4)	12.5 (10–20)

ACE: angiotensin-converting enzyme; ARB: angiotensin II receptor blocker; RAS: renin-angiotensin system.

**Table 3 jcm-11-02939-t003:** Patient scores at the last follow-up, expressed as mean and interquartile ranges.

	Platinum-Based			Taxanes		
Variables	No RAS Inhibitor (*n* = 52)	RAS Inhibitor (*n* = 23)	*p*-Values	No RAS Inhibitor (*n* = 29)	RAS Inhibitor (*n* = 8)	*p*-Values
NCI-CTC 4.0	0.7 (0–1)	0.4 (0–1)	0.047 *	0.8 (0–1)	0.9 (0–1.3)	0.853
NPSI	6.8 (0–12.3)	4.3 (0–7.5)	0.225	6.3 (0–8)	15 (0–25.3)	0.734
CIPN-RODS	79.4 (72.8–94)	76.7 (68.5–84)	0.503	79 (70–94)	73.5 (69.8–76)	0.072
EQ5D5L	8.7 (6–9.3)	8 (6–9)	0.337	8.7 (6–10)	8.8 (8–9.3)	0.456
ESC (µS)						
Hands	63.3 (58–79)	66.7 (60–80)	0.508	68.7 (65.5–78.3)	72.6 (71.5–80.5)	0.132
Feet	65.5 (52–81)	71.1 (61–86)	0.289	74.2 (70.5–81.8)	67.6 (57.3–84.5)	0.670

Quantitative variables are expressed as mean (interquartile range); *p*-values were calculated with the Mann–Whitney U-test or Student’s *t*-test for quantitative variables, as appropriate. * *p* < 0.5 in RAS inhibitor vs. no RAS inhibitor groups. CIPN-RODS: Rasch-built overall disability scale for patients with chemotherapy-induced peripheral neuropathy; EQ5D5L: 5-level EuroQol Research Foundation questionnaire; ESC: electrochemical skin conductance; NCI-CTC 4.0: National Cancer Institute common toxicity criteria; NPSI: neuropathic pain symptom inventory; RAS: renin-angiotensin system.

## Data Availability

Not applicable.

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
