# Peer review of "Renin-Angiotensin-System Inhibitors for the Prevention of Chemotherapy-Induced Peripheral Neuropathy: OncoToxSRA, a Preliminary Cohort Study"

_jcm, 2022, doi:10.3390/jcm11102939_

Round 1

Reviewer 1 Report

The manuscript „Renin-angiotensin-system inhibitors for prevention of chemo-2 therapy-induced peripheral neuropathy: OncoToxSRA, a preliminary cohort study” is well written and clear. It is the first prospective study to investigate the effect of RAS inhibitors on peripheral neuropathy. The provided results are quite interesting for the community. Even though the effect is not strong, it provides first evidence that there is a potential neuroprotective effect for patients receiving platinum-based chemotherapy, which needs to be evaluated in future trials.

The following comments, however, need to be addressed:

  • The Material and methods section lacks information about the power calculation of the trial. Thus, the authors should add more information about the biometry and calculation of the number of cases necessary.
  • The Asterix in Figure 1 needs to be explained in the figure legend. However, the remark “103 completed follow-up* is not essential for the figure and, thus, could also be deleted.
  • BMI and Age are significantly different between the two study groups. However, BMI and Age are also associated with CIPN (Bao et al. 2016, Greenlee et al. 2017, Timmins et al 2022, Bulls et al. 2019, Petrovchich et al. 2019). Thus, a multivariate model should be added to the analysis in order to provide information about the combined effect.

Reviewer 2 Report

Frachet et al wrote an interesting paper entitled "Renin-angiotensin-system inhibitors for prevention of chemotherapy-induced peripheral neuropathy: OncoToxSRA, a preliminary cohort study". They designed an observational study on one hundred and twenty patients. In my opinion, the study is very well designed and the manuscript is well written.

However, I would like to ask you to add details related to blood pressure in the designed groups.

Moreover please discuss a bit role of chemotherapy on blood pressure e.g. (https://doi.org/10.1016/j.jash.2018.03.008)  . Please also discuss the role of hypertension in neuropathy e.g. (DOI: 10.1093/ajh/hpz058; DOI: https://doi.org/10.3122/jabfm.19.3.240).
